# A machine learning approach to explore individual risk factors for tuberculosis treatment non-adherence in Mukono district

**Haron W. Gichuhi**[1]*, **Mark Magumba**[2], **Manish Kumar**[3], **Roy William Mayega**[1]

**1** Department of Biostatistics and Epidemiology, Makerere University School of Public Health, Kampala, Uganda, **2** Department of Information Systems, Makerere University College of Computing, and Information Science, Kampala, Uganda, **3** Public Health Leadership Program, Gilling's School of Global Public Health, University of North Carolina, Chapel Hill, North Carolina, United States of America

* gichuhiaaron@gmail.com

**Data Availability Statement:** The data represented in the manuscript is sufficient. However, the raw data used for modeling can be accessed in the repository below. For further explanations about

## Abstract

Despite the availability and implementation of well-known efficacious interventions for tuberculosis treatment by the Ministry of Health, Uganda (MoH), treatment non-adherence persists. Moreover, identifying a specific tuberculosis patient at risk of treatment non-adherence is still a challenge. Thus, this retrospective study, based on a record review of 838 tuberculosis patients enrolled in six health facilities, presents, and discusses a machine learning approach to explore the individual risk factors predictive of tuberculosis treatment non-adherence in the Mukono district, Uganda. Five classification machine learning algorithms, logistic regression (LR), artificial neural networks (ANN), support vector machines (SVM), random forest (RF), and AdaBoost were trained, and evaluated by computing their accuracy, F1 score, precision, recall, and the area under the receiver operating curve (AUC) through the aid of a confusion matrix. Of the five developed and evaluated algorithms, SVM (91.28%) had the highest accuracy (AdaBoost, 91.05% performed better than SVM when AUC is considered as evaluation parameter). Looking at all five evaluation parameters globally, AdaBoost is quite on par with SVM. Individual risk factors predictive of non-adherence included tuberculosis type, GeneXpert results, sub-country, antiretroviral status, contacts below 5 years, health facility ownership, sputum test results at 2 months, treatment supporter, cotrimoxazole preventive therapy (CPT) dapsone status, risk group, patient age, gender, middle and upper arm circumference, referral, positive sputum test at 5 and 6 months. Therefore, machine learning techniques, specifically classification types, can identify patient factors predictive of treatment non-adherence and accurately differentiate between adherent and non-adherent patients. Thus, tuberculosis program management should consider adopting the classification machine learning techniques evaluated in this study as a screening tool for identifying and targeting suited interventions to these patients.

## Introduction

Tuberculosis (TB), a curable and preventable infectious disease, remains a public health challenge globally, leading to serious economic and social consequences [1]. Although, curable and

the dataset, kindly reach out to me via gichuhiaaron [at] gmail [dot] com https://osf.io/ejahu/?view_only=9deca587fe1640d89ef4f058bdf0fc9e.

**Funding:** The authors received no specific funding for this work.

**Competing interests:** The authors have declared that no competing interests exist.

preventable, with well-known treatment strategies, its treatment management is challenging. Of these challenges, its treatment non-adherence and the consequent difficulty in identifying these non-adherent patients has been widely reported [2–5]. Moreover, tuberculosis treatment non-adherence is reported to reduce treatment success, increasing the risk of the patients developing drug-resistant strains, burdening the health systems [6], increasing family financial hardships [7], and spreading tuberculosis not only in communities within the country but also globally due to its infectious nature. Certainly, reasons for tuberculosis treatment non-adherence are complex and intertwined between the patient, health system & healthcare providers factors [8–11]. This in turn increases tuberculosis morbidity and mortality globally [1].

In Uganda, despite the availability and implementation of well-known efficacious interventions for tuberculosis prevention and treatment by the Ministry of Health, Uganda (MoH), tuberculosis programs still report high treatment non-adherence rates, with treatment success rate currently estimated at 72% [12]. These high treatment non-adherence rates possibly imply an existence of factors intrinsic to the patients or the treatment strategies amongst others. Indeed, several studies attempting to identify these patient factors aiming to avert high treatment non-adherence rates have been conducted [7,13–18]. These studies employed both traditional statistics and epidemiological approaches utilizing logistic regression and other generalized linear models. Although these studies identified some factors associated with tuberculosis treatment non-adherence, their applicability in identifying an individual tuberculosis patient at risk of treatment non-adherence is limited.

This is mainly because; a) these models are best for making inferences that are aimed at understanding the association between the predictors and the response and not for prediction [19], and b) Sometimes, the true relationship is more complicated in which case a linear model may not provide an accurate representation of the relationship between the input and output variables [20,21], c) the researchers do not split their data into training and testing sets and thus do not evaluate the resultant' models on raw datasets [21], d) the traditional statistics are "limited in handling highly dimensional and correlated variables (collinearity assumption)"[19], thus dropping some would be important variables from the resultant models.

In contrast, machine learning (ML), the creation of computer programs that can learn and therefore improve their performances by gathering more data and experiences [21], could prove beneficial in exploring individual patient factors predictive of tuberculosis treatment non-adherence. In fact, several studies have reported the effectiveness of machine learning models in accurately illustrating the target parameters for implementing stakeholders to ensure adherence to tuberculosis treatment and other chronic diseases [4,13,22–25]. Yet, studies conducted in the Ugandan contexts have not adequately utilized machine learning as a method to generate patient predictors of tuberculosis treatment non-adherence [5,26,27]. Therefore, this study set out to explore machine learning algorithms to quantify patient factors predictive of tuberculosis treatment non-adherence that could potentially identify an individual patient at risk of tuberculosis treatment non-adherence.

## Previous machine learning related works in tuberculosis

Tuberculosis, due to its infectious nature, increasing antimicrobial resistance and prevalence in low-and-middle-income (LMIC) countries, for example, India [28], Pakistan [29] Afghanistan [30], Morocco [13,31], Kenya [32,33], and Uganda [3,34–37] has been widely studied. Characteristically, these previous studies gathered demographic and medical histories of a cohort and observed their adherence and outcomes. The researchers then retrospectively applied machine learning algorithms e.g., logistic regression [13,29,30], support vector machine [29], and random forest [29], to determine variables predictive of treatment failure or non-adherence.

For instance, a study carried out in Iran [38] seeking to evaluate and compare different machine learning methods to predict the outcome of the tuberculosis treatment course, used a training dataset (N = 4515) and testing dataset (N = 1935) to explore six machine learning algorithms, namely: decision trees (DT), artificial neural network (ANN), logistic regression (LR), radial basis function (RBF), Bayesian networks (BN), and support vector machine (SVM).

To evaluate the algorithms, the Iranian investigators computed the prediction accuracy, F-measure, and recall metrics. They found out that decision trees (C4.5) performed the best with model fitness and prediction accuracy of 84.45% and 74.21% respectively. These study findings were like studies conducted in Kenya [39] and China [40] that equally reported the decision trees prediction to perform better with an accuracy of 90% and 70.9% respectively. Noteworthy, the study from Kenya concluded that machine learning techniques have the potential to identify patients at risk for viral failure before their scheduled measurements.

In another study [41] carried out in the USA to detect whether a patient will experience an adverse event due to coronary artery disease (CAD) within a 10-year time frame. Here, the researchers collected 21,460 patients' records. Of these, 75% were used for training and 25% for validation. They trained Logistic Regression, Random Forest, Boosted Trees, CART, and Optimal Classification Trees (OCT) classifiers. After evaluation, they found out that random forest was able to identify specific patients with an accuracy of 84.29% closely followed by OCT at 81.54%.

Newer work has explored the utility of advanced machine learning techniques such as support vector machines (SVM), Artificial Neural Networks (ANN), and more for improved classification accuracy. A study by Mian et al., [29] applied SVM in a dataset containing 275 pulmonary tuberculosis symptomatic and confirmed multidrug-resistant (MDR) cases age > = 15 with no gender discrimination for feature selection (FS) algorithms to identify and diagnose MDR tuberculosis in Pakistan. The researchers built and evaluated seven classifiers: random forest, k-nearest neighbors, support vector machine, logistic regression, least absolute shrinkage, selection operator (LASSO), artificial neural networks (ANNs), and decision trees. They found out that the two best-performing algorithms were SVM and RF with an accuracy of 78% and 74% respectively for patients' classification.

Further, previous work in this domain has shown that classification machine learning algorithms such as: decision trees (DT), random forests (RF), and support vector machines (SVM) perform better in other countries. Unfortunately, based on our literature search, we did not identify a study utilizing them for tuberculosis treatment non-adherence prediction in the Ugandan context. Thus, our study explored DT, RF, and SVM techniques amongst others.

Lastly, this study notes the increasing use of machine learning for specific customer loan default predictions in the banking sector [42], multilingual tweets classification for disease surveillance [43], and predicting an individual HIV/AIDs patient likely not to adhere to treatment [44]. Similarly, this study set out to replicate this in tuberculosis management and research.

## Materials and methods

### Study design and setting

Our retrospective study aimed at developing, evaluating, and recommending probable classification machine learning methods for the identification of important variables predictive of tuberculosis treatment non-adherence in the Mukono district. Our study defined tuberculosis treatment non-adherence as a surrogate measure as follows; "any patient who did not visit the health facility for a treatment drug refill in 28 days as was instructed by the health worker". This was abstracted as recorded in the health facility registers.

The study site, Mukono district, is in the central region of Uganda, with a population of 720,100 per the 2021 country statistical abstract [45]. The district is divided into four counties with 590 villages composed of three categories of populations: urban, peri-urban, and rural. However, the main economic activities within the district are farming, fishing and small-scale businesses. Within the district, six high-volume (treating > 100 patients) health facilities owned either by the government or private-not-for-profit (PFNP), serving urban, peri-urban, and rural populations were selected. These were 1) Kyetume Health Centre III (PNFP), 2) Mukono General Hospital (Government), 3) Naggalama Hospital (PNFP), 4) Mukono COU Hospital (PNFP), 5) Kojja Health Centre IV(Government), 6) Nakifuma Health Centre III (Government). We chose these health facilities as our study sites for several reasons. First, they treated >10 tuberculosis patients/per month. Second, they had a tuberculosis treatment success rate of < 75% from 1st January 2019—to 31st December 2021. Third, they had both outpatient (OPD) and inpatient (IPD) clinics. Finally, they are all located within the Mukono district.

## Data description

Our goal was to use a machine learning technique to identify important variables from a dataset comprising of 42 patient demographics and clinical characteristics that may predict treatment outcomes. The data was obtained from the routinely, longitudinal health facility registers, standardized by the ministry of health for the period starting 1st January 2019 to 31st December 2021. This data was abstracted using an electronic data capture screen developed using the Kobo Collect (v2021.3.4) platform. Kobo Collect is open-source software with both online and offline electronic data capture capabilities [46]. The tool was pre-tested before rolling out to the study.

A common practice in machine learning experimentations is to collect and utilize large volumes/quantities of data [47]. However, the number of data records required for a machine learning algorithm depends on the complexity of the model, the diversity of the data, and the performance goals of the model [21].

In general, the more data you have, the better the model will be able to learn and make accurate predictions. When developing prediction models for binary or time-to-event outcomes, a well-known rule of thumb for the required sample size is to ensure at least 10 events for each predictor parameter [47]. Thus, we estimated our sample size as follows. Total variables captured in the tuberculosis register = 42. But one of the variables is the outcome variable, thus; (42–1) * 10 = 410 records. However, cognizant of the limitations in existing health facility data–incomplete records, missing registers–in our study sites, we anticipated that to meet the minimum sample size computed above, abstracting records > 1000 would help us remain with a substantially high number of records enough for both training and testing our models even after data cleaning and removal of possible duplicate records. Therefore, we abstracted >1000 records from the drug susceptible pulmonary tuberculosis treatment units at the study sites.

Further, the data for each variable was extracted as it was presented in the registers. However, we anticipated the possibility of duplicates because of several reasons. This includes lack of generally agreed patient unique identifier, recording errors, and seeking treatment from several sources within the Mukono district.

## Patient eligibility

Through the tool, we abstracted patient records from the drug-susceptible pulmonary tuberculosis treatment units at the study sites. These were the patients who visited the study site with a

primary diagnosis of tuberculosis. They comprised both male and female patients, aged 18 years and above, who received and completed drug-susceptible pulmonary tuberculosis treatment at the study sites for the period starting 1st January 2019 to 31st December 2021 as captured in the 2019 health facility tuberculosis register. The 2019 health facility register is a newer, updated registers since the previous registers (2018 and below) did not capture some patient and clinical characteristics hence their use was discontinued countrywide. To meet the study objectives, the scope and ensure data of good quality, we identified the tuberculosis patients using the inclusion criteria; 1) All patients initiated on treatment for drug-susceptible pulmonary tuberculosis, who are male or female aged > 18 years. We considered patients aged 18 years and above because younger patients (< 18 years) do not make decisions on where and when to seek health care. 2) All patients' records with complete drug-susceptible pulmonary tuberculosis treatment outcomes are correctly filled in and meet the criteria 1. 3) All patients who received the drug-susceptible pulmonary tuberculosis treatment at the study site in the period of 1st January 2019 to 31st December 2021. We excluded all patients with incomplete drug-susceptible pulmonary tuberculosis treatment outcomes filled in and all patients who initiated treatment at the study sites but were later transferred out. The data for each variable was extracted as it was presented in the standardized health facility registers. The dependent variable was the tuberculosis treatment outcome. The independent variables included the patient's age, sex, category, disease classification, treatment drugs, and risk group among others.

## Data analysis

The data cleaning and basic descriptive statistics were done in STATA v15. During data cleaning, we transformed the categorical data into numerical data using numerical value labels aided by an already-defined codebook. This process was repeated for all other categorical variables within the dataset. For instance, the outcome variable was abstracted as either; cured, treatment completed, treatment failure, lost to follow-up, not evaluated, or died. However, using the codebook and the study definition of treatment non-adherence, we transformed this into two categories with labels "1" for adherent and "2" for non-adherent patients respectively.

## Prediction tools and software

R studio version 2022.02.3 Build 492 and, R version 4.2.1 were utilized for modeling and building the machine learning classification algorithms. Both are freely available data analytics software. Whereas R is an open-source, statistical, and data-centric programming language, R studio is an open-source integrated development environment (IDE), with a simple graphical user interface (UI). Furthermore, R studio provides a user-friendly point-and-click graphical user interface for the R programming language.

R provides many different algorithms for data mining and machine learning with flexible facilities for scripting experiments. We utilized the commonly used data processing libraries for data visualization, formatting, slicing, and conversion such as "*gtsummary*", "*tidyverse*", and "*dplyr*". Thereafter, we converted all the character variables into factor formats in preparation for statistical modeling. Finally, we formulated and added labels to the variables, generated a basic descriptive summary table using the "*gtsummary*" library, and saved the prepared data ready for applying machine learning algorithms.

We visualized the data using basic descriptive statistics to check for incompleteness, inconsistency, and inaccuracy (errors or outliers). The analysis and modeling results were presented using tables and graphs. Lastly, the resultant clean dataset obtained after analysis was converted, exported, and stored in password-protected comma-separated values (CSV) file, before its utilization for machine learning modeling purposes.

## Missing data handling

Various approaches to manage missing data exist. In this study, after the data cleaning process, we loaded the cleaned dataset in STATA to check and identify the patient demographics and clinical features columns that had any missing data values. From our dataset exploration, we identified that the laboratory tests had the most missing variables. Further investigation revealed that the identified missing laboratory tests were not carried out and thus were not recorded. Because we could not use the commonly used statistical computing techniques to estimate their respective values, we therefore, generated a new variable called "None" and replaced all the missing laboratory test values.

## Feature selection

Feature selection (FS) involves searching through all the attributes in the data to remove non-informative or redundant attributes whilst finding the subset of attributes that maximizes performance. Also, FS reduces overfitting (less opportunity to decide based on noise), improves accuracy (removing misleading variables), and reduces the training time. Due to the clinical nature of our dataset, and the research objectives, a two-stage feature selection procedure was followed. First, in building the algorithm and second, in the selection of feature importance based on the best-performing model.

In algorithm building, we applied the attribute evaluator and the filter ranker search method. In the filter ranker method, an attribute evaluator assigned a relevance score to each feature in the dataset. Thereafter, the attributes were ranked according to their relevance score in descending order. The features with a high score were selected and low-scoring features were eliminated during modeling.

## Building the machine learning algorithms

This was done in R-Studio, using the library **"caret"**, a widely used R machine learning package. We began by splitting our dataset into two using a ratio of 80:20, 80% for training, and 20% for testing the model respectively. Next, using the training dataset (80%), we built the five algorithms support vector machines, AdaBoost, artificial neural networks [48], decision trees [21], and Logistic regression [49]. Similarly, testing for each developed algorithm was done using the testing dataset.

For every algorithm developed, a 10-fold cross-validation was applied for model building with the training dataset. Internally, the training dataset was split into two with 90% of the data used for training and 10% used for testing. This was repeated 10 times before the final model was built from the training dataset.

The final model was tested using the 20% of the original dataset that was split during the 80:20 ratio and reserved for model testing. Using this testing dataset, we computed a confusion matrix to measure the accuracy (based on a 95% level of confidence), positive predictive value, negative predictive value, sensitivity, specificity for every machine learning model.

## Evaluation of the developed models

For the model performance evaluation, k-fold cross-validation was used. Cross-validation is a set of methods for measuring the performance of a given predictive model on new test data sets [48]. The rationale in cross-validation techniques is to divide the data into two sets, training sets–used to train (build) the model and the testing set (validation set)–used to test the model by computing the prediction error.

We implemented the repeated *k*-fold cross-validation method, whereby we randomly split our dataset into *k* sets. In this method, we split our data into 10-fold equal datasets. We used

9-fold (90%) datasets for training the model and the rest 1-fold (10%) of the dataset to evaluate the model performance. Thereafter, we evaluated the developed model with the test dataset (20%) to check its validity and accuracy using the unseen observations. From the results, we quantified the prediction error as the mean squared difference between the observed and the predicted outcome values.

## Performance measure of the developed model

Several methods to quantify the performances of the machine learning models exist. These include accuracy, F1 score, precision, recall, and Receiver Operating Curve (ROC). Accuracy measures how many observations, both positive and negative, were correctly classified by the algorithm. Accuracy is commonly used in a balanced classification problem in which every class is equally important to the researcher. Further, precision aims to quantify the proportion of positive identifications that were correct while recall tries to what proportion of actual positives was identified correctly. On the other hand, the F1 score combines precision and recall into one metric by calculating the harmonic mean between those two. However, the F1 score is mostly used in finding out about the positive class.

Furthermore, the receiver operating curve (ROC) is a chart that visualizes the tradeoff between the true positive rate (TPR) and false positive rate (FPR). Resultantly, classifiers with curves that are more top-left-side are better. i.e., the higher TPR and the lower FPR for each threshold the better. Yet, accuracy aids in quantifying the fraction of predictions our model got right incorporating both precision and recall. Therefore, we choose to utilize a confusion matrix to compute the model accuracy and use it as our best performance measure. The confusion matrix was computed the using the formula below.

$$accuracy = \frac{TP + TN}{TP + TN + FP + FN}$$

Where, true positives (TP) are the predicted values correctly predicted as actual positives. false positives (FP) are the predicted values incorrectly predicted as actual positives. i.e., negative values predicted as positive. Similarly, false negatives (FN) are the positive values predicted as negative. Whereas true negatives (TN) are the predicted values correctly predicted as actual negatives. Thus, we selected the best-performing model by using a criterion of the higher the computed accuracy, the better the performance of the developed model. The results are tabulated in Table 2.

## SHapley Additive exPlanations (SHAP)

SHapley Additive exPlanations (SHAP) is a "unified framework for interpreting machine learning models that assign each feature an importance value for a particular prediction" [50]. The rationale behind the SHAP is that for complex models e.g., ensemble methods or deep networks, a *simpler explanation model* defined as an interpretable approximation of the original model exists. To identify the critical variables, we first selected the best-performing model. Thereafter, we applied the SHapley Additive exPlanations (SHAP) to it.

Mathematically, SHAP applies sampling approximations to model training by assigning an importance value to each feature that represents the effect on the model prediction of including that feature. Finally, it computes the approximations of the effect of removing a given variable from the model by summing up the preceding differences for all possible subsets of all features. This is illustrated in the formula below.

$$\emptyset_i = \sum_{S \subseteq F\{i\}} \frac{|S|!(|F| - |S| - 1)!}{|F|!} \left[ f_{s \cup \{i\}}\left(x_{s \cup \{i\}}\right) - (f_s)x_s \right]$$

Where, $F$ is the set of all features, $S \subseteq F$ is the subset of all features, $f_{s \cup \{i\}}$ is the model trained with a given feature ($i$) present, while $f_s$ is another model trained with the feature withheld. $x_s$ represents the values of the input features in the set $S$.

Further, the SHAP values attribute to each feature the change in the expected model prediction when conditioning on that feature. From these attributions, SHAP thus explains how to get from the base value $E[f(z)]$ that would be predicted if we did not know any features of the current $f(x)$. This uses classic equations from cooperative game theory to aid in interpreting the black-box machine-learning models by computing the explanations of the model predictions.

Therefore, by applying SHAP, we were able to rank our predictive features according to their sign and magnitude in response to their contribution to the outcome. Based on the feature's contribution sign (whether positive or negative), and the magnitude, the critical predictors for either adherence or non-adherence were identified. The results were presented graphically.

## Ethical clearance

Ethical approval was sought and granted (approval protocol number 047) by the institutional review board (IRB) at Makerere University School of Public Health to conduct the research. In addition to the approval, we were granted permission by the Mukono district health officer (DHO) to carry out the study in the six (6) study sites. Further administrative clearance from the health facilities in-charges and the director at Mukono General Hospital and Mukono COU hospital was sought and granted.

## Results

In total, 1,004 tuberculosis patient records were abstracted. After data cleaning, 166 records were eliminated because they did not meet the inclusion criteria (age = > 18 years). Therefore, 838 records were considered for analysis and modeling. The records belonged to three (50%) government hospitals; Mukono general hospital (252), Kojja health center IV (116) and Nakifuma health center III (79) and, three (50%) private not-for-profit (PFNP) hospitals namely, Mukono COU hospital (132), Naggalama hospital (156) and Kyetume health center III (103) as shown in Fig 1 below.

The patient sex (either "male" or "female") was balanced (57% male) as shown in Fig 2. The patient's mean age was 38.3 years with a standard deviation (SD) of 13.7 years. Similarly, the patient's mean weight and standard deviation were 42.0 and 20.1 respectively. Out of all patients, 568 had contacts > 5 years (mean of 1.6 and a standard deviation of 2.2), while 340 patients had contacts < = 5 years, with a mean and standard deviation of 0.7 and 1.6 respectively.

However, the categorical treatment outcome was unbalanced with 681 (81.2%) and 157 (18.8%) belonging to adherent and non-adherent classes respectively. Of all the tuberculosis tests done, most of the patients had GeneXpert tests (618), closely followed by smear microscopy (502) and tuberculosis LAM (39). 112 patients had no laboratory test done. Detailed background of the study participants is presented in Table 1 below.

From the above findings, SVM had the highest accuracy, 91.28%, (AdaBoost performed better than SVM when ROC is considered as evaluation parameter). Looking at all five evaluation parameters globally, AdaBoost is quite on par with SVM. Both Artificial neural networks (ANN) and Logistic regression had equal accuracy of 88.30%. However, Random Forest had the lowest F1 score of 0.665 compared to other algorithms that scored greater than 0.800.

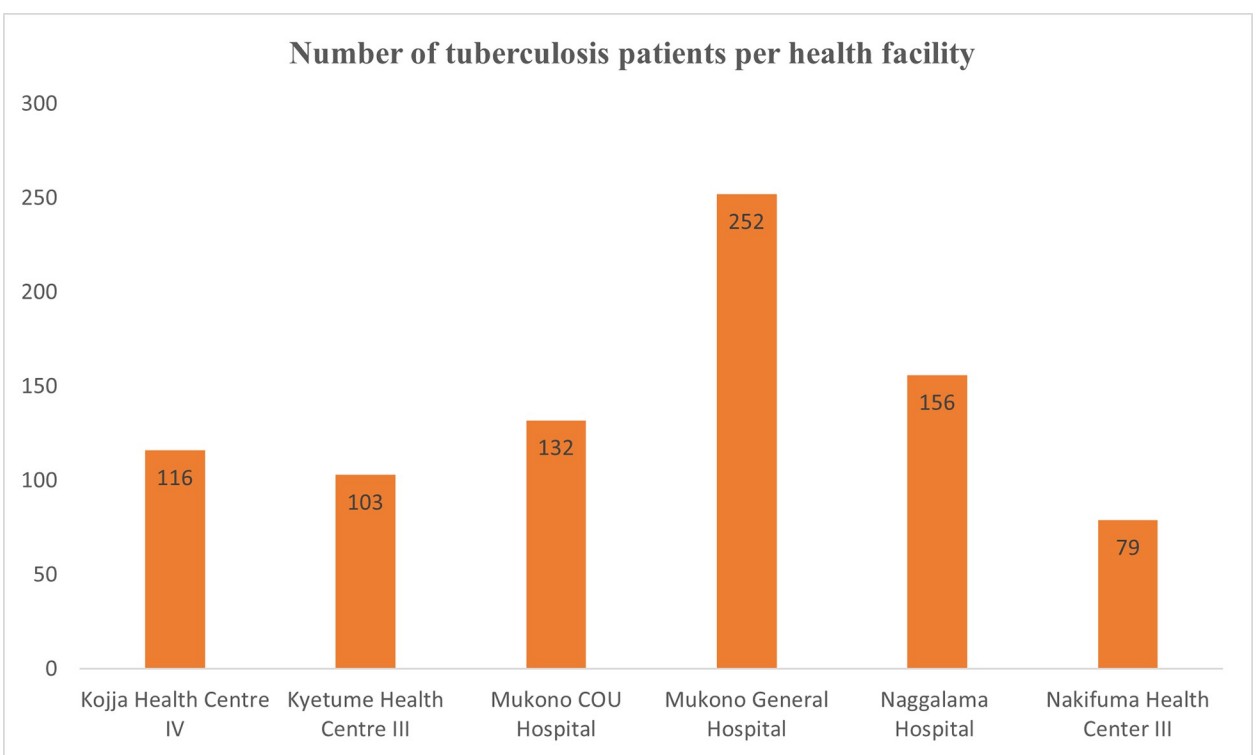

**Fig 1. The frequency distribution of tuberculosis patients per health facility in the tuberculosis data.**

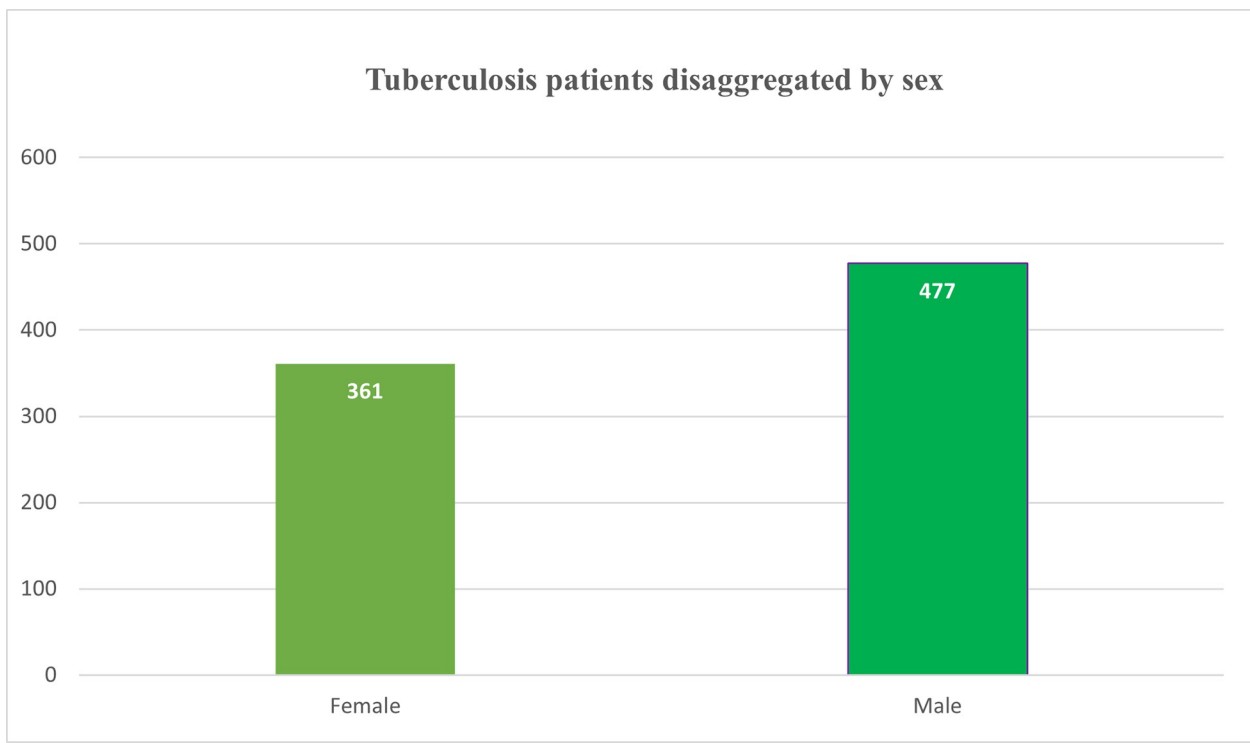

**Fig 2. Frequency of patient sex in the tuberculosis data.**

**Table 1. Summary statistics for key predictor and outcome variables.**

| Characteristic | Overall, N = 838[1] | adherent, N = 681[1] | non-adherent, N = 157[1] |
|---|---|---|---|
| **Health Facility Ownership** | | | |
| Government | 447 (53%) | 373 (55%) | 74 (47%) |
| Private not-for-profit (PFNP) | 391 (47%) | 308 (45%) | 83 (53%) |
| **Sex** | | | |
| Female | 360 (43%) | 306 (45%) | 54 (34%) |
| Male | 478 (57%) | 375 (55%) | 103 (66%) |
| **Patient Category** | | | |
| Foreigner | 3 (0.4%) | 3 (0.4%) | 0 (0%) |
| National | 834 (100%) | 677 (99%) | 157 (100%) |
| Refugee | 1 (0.1%) | 1 (0.1%) | 0 (0%) |
| **Disease Classification** | | | |
| Extrapulmonary (EPTUBERCULOSIS) | 28 (3.3%) | 23 (3.4%) | 5 (3.2%) |
| Pulmonary Bacteriologically confirmed (PBC) | 507 (61%) | 410 (60%) | 97 (62%) |
| Pulmonary Clinically Diagnosed (PCD) | 303 (36%) | 248 (36%) | 55 (35%) |
| **TUBERCULOSIS Type** | | | |
| New Case (N) | 768 (92%) | 626 (92%) | 142 (90%) |
| Relapse (R) | 39 (4.7%) | 32 (4.7%) | 7 (4.5%) |
| Treatment after failure (TF) | 20 (2.4%) | 12 (1.8%) | 8 (5.1%) |
| Treatment after loss to follow-up (TL) | 11 (1.3%) | 11 (1.6%) | 0 (0%) |
| **Regimen** | | | |
| 2RHZE/10RH | 8 (1.0%) | 4 (0.6%) | 4 (2.5%) |
| 2RHZE/4RH | 825 (98%) | 673 (99%) | 152 (97%) |
| Others | 5 (0.6%) | 4 (0.6%) | 1 (0.6%) |
| **Transfer In** | 57 (6.8%) | 48 (7.0%) | 9 (5.7%) |
| **Referral** | | | |
| Community | 193 (36%) | 167 (39%) | 26 (25%) |
| Facility | 344 (64%) | 264 (61%) | 80 (75%) |
| Unknown | 301 | 250 | 51 |
| **HIV status** | | | |
| Known HIV positive at TUBERCULOSIS diagnosis | 231 (28%) | 190 (28%) | 41 (26%) |
| Negative | 422 (50%) | 347 (51%) | 75 (48%) |
| Newly tested HIV positive at TUBERCULOSIS diagnosis | 179 (21%) | 140 (21%) | 39 (25%) |
| Unknown HIV status | 6 (0.7%) | 4 (0.6%) | 2 (1.3%) |
| **DOT Model** | | | |
| Digital Community DOT | 234 (28%) | 202 (30%) | 32 (20%) |
| Facility DOT | 126 (15%) | 98 (14%) | 28 (18%) |
| Non-Digital Community DOT | 443 (53%) | 358 (53%) | 85 (54%) |
| None | 35 (4.2%) | 23 (3.4%) | 12 (7.6%) |
| **Treatment Supporter** | | | |
| Community Volunteer | 14 (1.7%) | 12 (1.8%) | 2 (1.3%) |
| Family Member | 414 (49%) | 336 (49%) | 78 (50%) |
| Health Worker | 5 (0.6%) | 4 (0.6%) | 1 (0.6%) |
| None | 405 (48%) | 329 (48%) | 76 (48%) |

We built and evaluated 5 (five) different machine-learning models based on the resultant dataset. Thereafter, we computed the metrics (precision, recall, accuracy, receiver operating curve (ROC), and F1 score) of these developed models as shown in the table below.

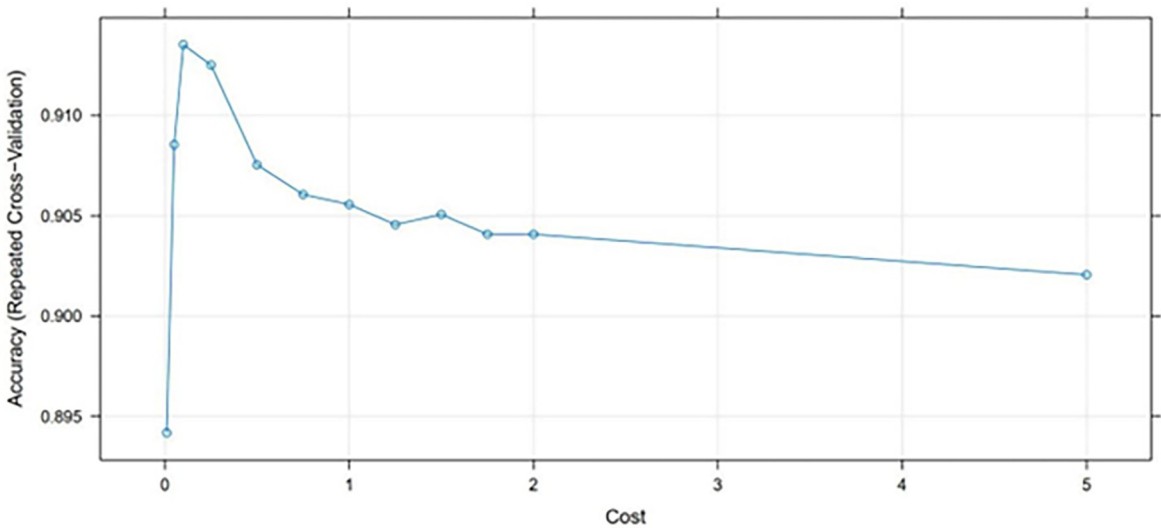

**Fig 3. Accuracy versus the cost of the support vector machine model.**

Notably, we tuned the SVM classifier to get better performance with different values of C (cost) per kernel function. With the linear kernel function, we obtained the best model accuracy of 91.28% with a cost parameter of c = 0.1, as shown in Fig 3 below. Further exploration revealed that the model accuracy decreased with increased cost.

Out of the five, the support vector machine (SVM) model had the highest accuracy of 91.28%. Thus, we applied the Shapley Additive exPlanations (SHAP) to this model. This was to identify the important variables that could estimate the tuberculosis treatment non-adherence. The SHAP ranked all the variables used for modeling based on their contribution toward the tuberculosis treatment outcome. These are shown in Fig 4 below.

On further exploration, the SHAP identified predictors' importance based on their magnitude and sign (positive and negative) in relation to their contribution. The positive predictors were associated with adhering, while negative predictors are associated with non-adherence (Fig 5).

From the graphical representation above, disease classification, drug regimen, patient category, DOT model, transfer-in, HIV status, contacts over 5 years, weight, and smear microscopy results were identified to contribute positively towards treatment adherence.

In contrast, tuberculosis type, GeneXpert results, sub-country, anti-retroviral status, contacts below 5 years, health facility ownership, sputum test results at 2 months, treatment supporter, CPT Dapsone status, risk group, patient age, gender, middle and upper arm circumference (MUAC), referral, positive sputum test at 5 months and 6 months were indicative of treatment non-adherence.

## Discussion

Our support vector machine (SVM) model yielded a high accuracy (91.28%) and high recall (0.9130) compared to other ML algorithms (Table 2 in the results section) in classifying treatment non-adherence. However, the high recall attained by the SVM model was expected. This is due to the internal workings of the SVM algorithm that aims to identify a hyperplane that separates the training data, by maximizing the gap between these hyperplanes so that in case of any new data, it's mapped with a maximum gap separating the classes.

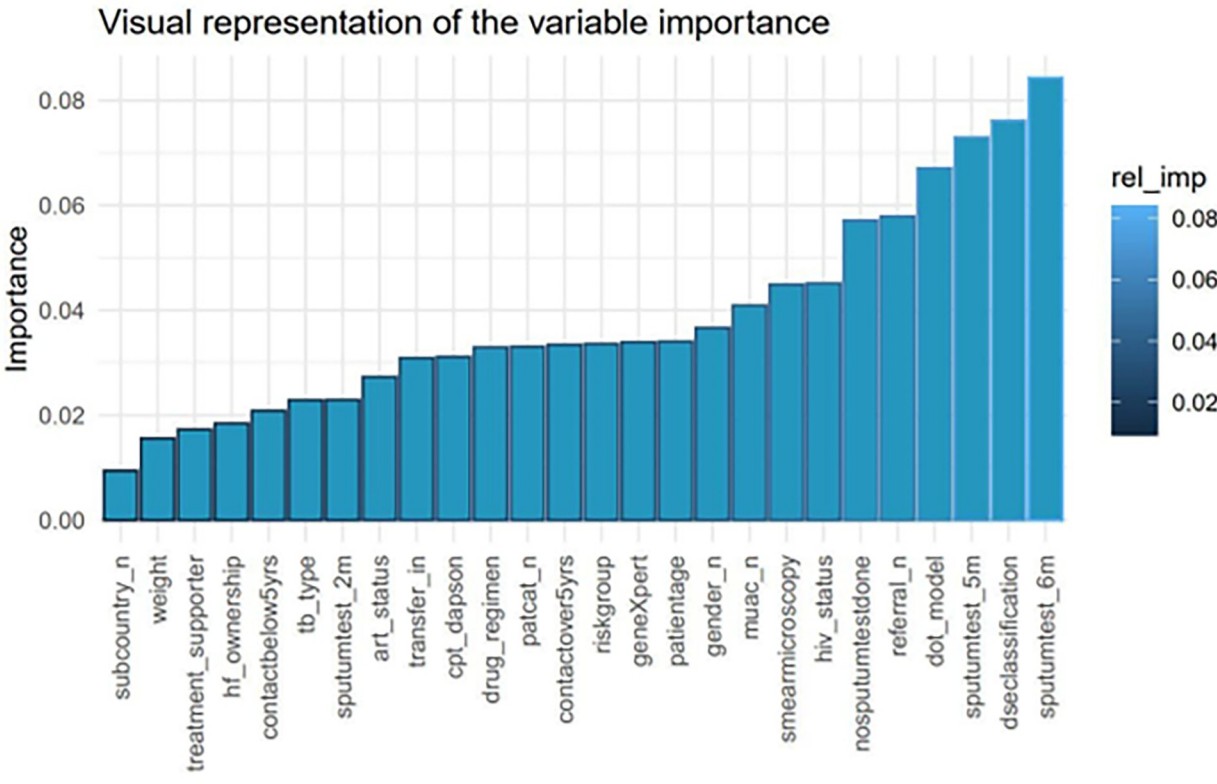

**Fig 4. Visualization of the independent variables' importance using the SHAP.**

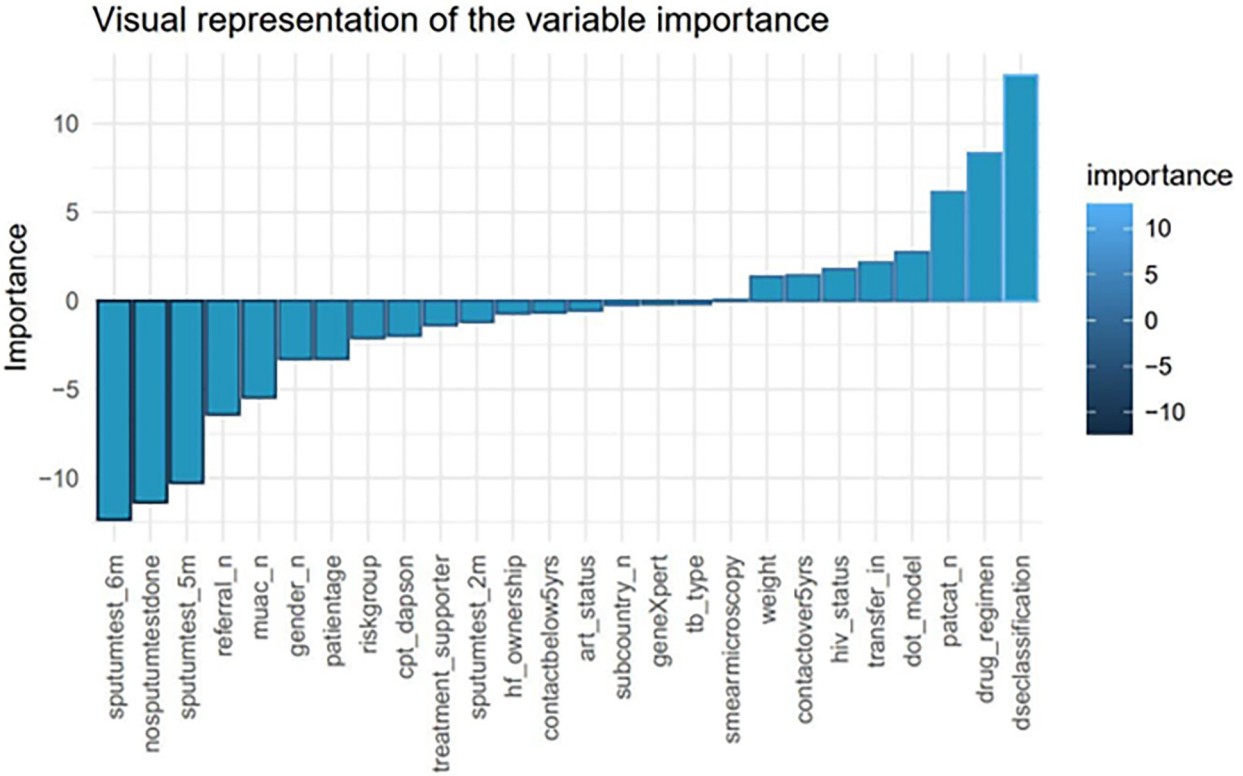

**Fig 5. Visual representation of the positive and negative predictors.**

**Table 2. Evaluation results of the developed models.**

| No. | Prediction algorithm | Evaluation metric | | | | |
|---|---|---|---|---|---|---|
| | | Precision | Recall | F1score | ROC | Accuracy (%) |
| 1. | Support vector Machine (SVM) | 0.916 | 0.913 | 0.914 | 0.870 | 91.28 |
| 2. | AdaBoost | 0.913 | 0.911 | 0.911 | 0.920 | 91.05 |
| 3. | Random Forest (RF) | 0.898 | 0.900 | 0.665 | 0.929 | 89.97 |
| 4. | Logistic regression | 0.885 | 0.883 | 0.884 | 0.896 | 88.30 |
| 5. | Artificial Neural Networks (ANN) | 0.879 | 0.883 | 0.881 | 0.902 | 88.30 |

Indeed, these characteristics exhibited by SVM have led to its wide adoption and utilization in several studies [22,29] that aimed to predict treatment failure and risk factors in tuberculosis and other illnesses. In all these studies, SVM algorithms were reported to have attained high accuracy and recall like ours. Thus, our findings of SVM as the best performer further confirm its robustness as a classification algorithm.

Noteworthy, all the five machine learning algorithms investigated were able to discriminate between the outcome class with high precision ($> 0.87$) considering all the 41 study features. These results indicate that classification machine learning models can be used to map the different predictors to their respective classes. Likewise, these capabilities have been reported in similar rather challenging scenarios like predicting viral failure [51], tweets classification for disease surveillance [43], identification of HIV predictors for screening [52], and cancer [53]. Therefore, this demonstrates the potential and applicability of machine learning algorithms to provide insights in scenarios where human decision-making would be limited.

Methodologically, our work was related to other studies predicting outcomes by inferring a model being trained by a set of historical data [15,22,25,40,54–56]. Given appropriate assumptions, such techniques allow for valid predictions about the counterfactual outcomes under different settings for determining interventions. However, the machine learning techniques require exact knowledge of intervention outcomes which should be clearly labeled.

Our study focused on developing and evaluating different machine learning algorithms to explore the risk factors predictive of treatment non-adherence. Thus, after training the models, we evaluated them for performance and accuracy. The metrics applied were precision (false positive), recall (false negative), F1 score, accuracy, and receiver operating curve (ROC). However, the application of these metrics in binary classifications is dependent on "when". For instance, while dealing with class-imbalanced datasets–a dataset for a classification problem in which the total number of labels of each class differs significantly- both precision and recall are the go-to metrics.

However, due to the clinical nature of our study, we were interested in a model with high recall. That is, we wanted a model capable of identifying individual patients, based on the demographics and clinical characteristics as captured in the health facility registers, who would not adhere to treatment, even if that meant having some false positives (patients wrongly identified as not adherent yet they are). Based on the imbalanced dataset obtained, this could have affected the F1 score of some of our algorithms leading to RF scoring the lowest (0.665) compared to other algorithms evaluated. This probably shows that RF is highly affected by dataset imbalance.

In a real clinical setting, false positives may lead to additional unnecessary clinical examinations, and extra laboratory tests and even frighten some patients. However, the public health benefits outweigh this setback. This is because the additional tests could help identify and take care of other unaware sick people thus reducing possible community spread, transmission, and possible death.

### Identification of patient characteristics for tuberculosis treatment non-adherence

A deeper exploration of the best performing SVM model identified some variables predictive of tuberculosis treatment non-adherence. We relied on the SHAP measurement for each independent variable contribution to the treatment outcome for the identification of these variables. From our findings, tuberculosis type, GeneXpert results, sub-country, anti-retroviral status, contacts below 5 years, health facility ownership, sputum test results at 2 months, treatment supporter, CPT dapsone status, risk group, patient age, gender, MUAC, referral, positive sputum test at 5 months and 6 months were predictive of treatment non-adherence. These results are like other studies conducted in lower- and middle-income countries exploring the patient characteristics associated with treatment non-adherence.

Notably, positive sputum tests at 5 months and 6 months were identified by the model as indicative of treatment non-adherence. However, when a patient comes into the health facility with a diagnosis, the health worker for example would not know whether the patient would do a sputum test at 5 or 6 months. Interestingly, the fact that we trained our model using retrospective data, indicates that our algorithm can fully categorize patient's adherence only after the sputum test at 5 or 6 months are conducted. Furthermore, this could potentially serve as an early warning to the health worker that, if the other non-adherence factors identified by the model are present, special guidance should be taken to also inform the patient on the importance of completing all the sputum tests and most especially during months 5 and 6.

Patient age and gender had equal importance in contributing to treatment non-adherence. This is possible because gender-based roles and responsibilities increase with the increase in age (16, 27). As men mature, they start engaging in income-generating activities that are not only demanding but also time-consuming. In turn, this may hinder men from going to health facilities to pick up their drug refills or even taking medicines as prescribed. Similarly, women also take up responsibilities like childbearing, and household chores that equally may hinder them from going for drug refills.

Various studies have reported on the significance of patient risk groups and residency as key determinants in treatment adherence [5,57]. In our modeling, we equally found the two characteristics to be predictive of whether a patient will adhere to treatment or not. Further, we found out that some patients who were not residents of our study site catchment areas attended the health facilities located within Mukono. This could be because of the stigma associated with tuberculosis as was discussed by [58]. In a similar scenario for covid-19 treatment seeking, a study by Muttamba et. al. . . [59] also cited stigma as a major hindrance. However, we noted that the grouping of risk groups as others could have masked other types of patient risk categories that could be on tuberculosis treatment and probably not adhering.

Whether a patient had a treatment supporter or not was found to be predictive of treatment non-adherence. Previous findings identified the important role played by treatment supporters in supporting chronic patients to adhere to treatment [60]. With the patient at liberty to choose between either a family member, community health worker or health worker as a supporter, the presence or lack of one thereof greatly impedes the treatment taking.

From our findings, both ART status and CPT Dapsone status were among the factors identified by our model. Certainly, CPT Dapsone is prescribed to HIV positive patients. However, HIV, like tuberculosis is a chronic disease with a high drug pill count. Evidence shows [2,36] that, the drug burden for these chronic and infectious diseases can overwhelm the patient who either decides to take drugs for one of the diseases and not take the others based on either drug reactions or side effects. However, newer studies should consider disaggregating the tuberculosis patient according to their HIV status for further modelling.

Middle and upper arm circumference (MUAC) measurements, an indicator of patient nutrition status was also found to be predictive of non-adherence. This concurred with two different studies [61,62] both conducted in Uganda, among tuberculosis patients, that found that undernutrition leads to unfavorable treatment outcomes. This finding also resonated with the fact that tuberculosis drugs are administered according to patient weight at diagnosis. However, upon treatment initiation, tuberculosis drugs have been shown to affect the patient's appetite.

## Limitations

Despite the study findings, we encountered some limitations. First, the use of health facility tuberculosis registers as a data source may not fully describe/capture the entire non-adherence process. This is partly because these registers leave out some social-environmental, social-demographic factors like patient occupation, marital status, number of children, and distance from the health facilities that have been shown by other qualitative studies to be associated with non-adherence. Secondly, our definition of tuberculosis treatment non-adherence as a surrogate measure does not do a good job in measuring the actual tuberculosis treatment non-adherence. However, we successfully identified some important factors that could guide non-adherence screening at the health facility despite the lack of environmental and social-demographic factors.

## Conclusion

Our study findings indicate that supervised machine learning algorithms can discriminate between adherent and non-adherent patients, with high accuracy and high recall. Thus, the resultant machine learning models can potentially be used as a regular simplistic tool to explore and identify individual risk factors for tuberculosis treatment non-adherence. This will complement the existing treatment adherence strategies by prioritizing the limited healthcare resources to the neediest patient thereby lowering the costs implications for active supervision and support for all tuberculosis patients.

Furthermore, we built all five classification machine learning models using the data collected routinely in a typical tuberculosis clinic in LMIC. On further experimentation with these datasets, using SHAP, we were able to identify some predictors of tuberculosis treatment non-adherence. Therefore, classification machine learning algorithms mainly; logistic regression, support vector machine (SVM), random forest (RF), AdaBoost, and artificial neural networks (ANN), can be built from the readily available health facility registers data.

Finally, whereas we front machine learning as an alternative advanced technique for finding hidden patterns in data, it comes at a cost. This cost is incurred in skillsets requirement, computing infrastructure, data collection, cleaning, labeling, wrangling, and modeling processes. Thus, this not only calls for preparations to incur the mentioned costs but also to invest time to experiment with different machine learning algorithms. This is to ensure that persons with the right skill set, computing infrastructure, and data of high quality can build machine learning models that can generate insights, to guide in both decision-making and policy formulations to better healthcare in our communities.

### Recommendations & future work

We suggest that future work should collect more data from a bigger population, to enrich the dataset for modeling treatment non-adherence. More data could be collected by additional of more study sites. Further, qualitative data obtained through administering key informant interviews to both the health workers and the patients could introduce more dimensionality to

the dataset. These qualitative data, though needing prior data transformations will help in better understanding the complex role of human behavior in treatment non-adherence. Also, it will greatly inform and supplement our findings to get a complete picture of the risk factors for tuberculosis treatment non-adherence.

Second, we recommend that tuberculosis health providers should prioritize educating the patients on the importance of refilling their treatment packages and the swallowing the medicines as advised by the health workers to minimize the risks and challenges associated with treatment non-adherence. Further attention/supervision should be given to any tuberculosis patients screened and identified to present with any of our identified factors during treatment.

Third, to the tuberculosis management programs, we recommend an additional field in the tuberculosis health facility registers to capture the actual risk group of the patient instead of employing the "others" option. Likewise, identified tuberculosis patients with several contacts > 5 years exposed to tuberculosis through household contact should be prioritized for screening so they can be started on treatment to prevent infection or progression of the disease.

Fourth, to the hospital management, frequent data quality assessments should be performed to address issues in missing data variable entries in the registers. Additionally, this will ensure that the management can address some issues like patients missing contact numbers or modalities of treatment support hence improving the quality of documentation.

Finally, researchers should not shy away from experimenting with machine learning techniques in addressing tuberculosis treatment refill management even in settings with limited data settings like Uganda and other lower-and-middle-income countries. This is because, these experimentations could reveal further insights and information regarding tuberculosis treatment non-adherence management to further augment what is already known.

## Acknowledgments

To the Mukono district health officer (DHO) who permitted us to conduct and collect data for this study in his district. Special gratitude to the management of Kyetume Health Centre III, Mukono General Hospital, Naggalama Hospital (PNFP), Mukono COU Hospital, Kojja Health Centre IV, and Nakifuma Health Centre III. Ivan Mwesigwa, the Mukono district biostatistician, and the respective health facilities data officers for the support rendered during the data abstraction process. To Dr. Vincent M. Kiberu, Dr. Alice Mugisha, Dr. John Ssenkusu and Dr. Ronald Muhumuza Kananura, who provided immeasurable insights and advice during the entire study period.

## Author Contributions

**Conceptualization:** Haron W. Gichuhi, Mark Magumba, Roy William Mayega.

**Data curation:** Haron W. Gichuhi.

**Formal analysis:** Haron W. Gichuhi.

**Investigation:** Haron W. Gichuhi.

**Methodology:** Haron W. Gichuhi, Mark Magumba, Manish Kumar, Roy William Mayega.

**Project administration:** Haron W. Gichuhi.

**Resources:** Haron W. Gichuhi.

**Software:** Haron W. Gichuhi.

**Supervision:** Haron W. Gichuhi, Mark Magumba, Manish Kumar, Roy William Mayega.

**Validation:** Haron W. Gichuhi, Mark Magumba.

**Visualization:** Haron W. Gichuhi.

**Writing – original draft:** Haron W. Gichuhi, Mark Magumba, Roy William Mayega.

**Writing – review & editing:** Haron W. Gichuhi, Mark Magumba, Manish Kumar, Roy William Mayega.

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
