## [Decision Letter · Decision Letter 0]

25 Jan 2023

PGPH-D-22-01989

A machine learning model to explore individual risk factors for tuberculosis treatment non-adherence in Mukono district

Dear Gichuhi,

Thank you for submitting your manuscript to PLOS Global Public Health. After careful consideration, we feel that it has merit but does not fully meet PLOS Global Public Health’s publication criteria as it currently stands. Therefore, we invite you to submit a revised version of the manuscript that addresses the points raised during the review process.

We look forward to receiving your revised manuscript.

Kind regards,

Collins Otieno Asweto, PhD

Academic Editor

Journal Requirements:

2. Please provide separate figure files in .tif or .eps format only and remove any figures embedded in your manuscript file. Please also ensure that all files are under our size limit of 10MB.

3. We have noticed that you have cited Table 5 in the manuscript file but there are no corresponding tables in the manuscript. Please amend your manuscript to include this table, noting that tables should not be uploaded as individual files.

Reviewers' comments:

Reviewer's Responses to Questions

**Comments to the Author**

1. Does this manuscript meet PLOS Global Public Health’s publication criteria? Is the manuscript technically sound, and do the data support the conclusions? The manuscript must describe methodologically and ethically rigorous research with conclusions that are appropriately drawn based on the data presented.

Reviewer #1: Yes

Reviewer #2: Partly

2. Has the statistical analysis been performed appropriately and rigorously?

Reviewer #1: Yes

Reviewer #2: Yes

3. Have the authors made all data underlying the findings in their manuscript fully available (please refer to the Data Availability Statement at the start of the manuscript PDF file)?

Reviewer #1: Yes

Reviewer #2: Yes

4. Is the manuscript presented in an intelligible fashion and written in standard English?

Reviewer #1: Yes

Reviewer #2: Yes

5. Review Comments to the Author

Reviewer #1: Excellent paper. Well written, clear and innovative.

I have a concern as to your choice of the SVM model on one hand and Accuracy on the other hand. 5 models were used to train your AI algorithm and 5 performance matrix were assessed. The following conclusion seems to misrepresent the outcomes unless I am missing something:

"Of the five developed and evaluated models, SVM performed the best with an accuracy of 91.28% compared to RF (89.97%), LR (88.30%), ANN (88.30%), and AdaBoost (91.05%) respectively."

I would say of the five models evaluated, SVM had the highest accuracy (AdaBoost performed better than SVM when AUC is considered as evaluation parameter). Looking at all 5 evaluation parameters globally, AdaBoost is quite on par with SVM.

The different evaluation parameters tell different stories. It may be good to explain clearly if one parameter is classed above the others (in this case, your abstract singles out Accuracy to mention superiority of the SVM model) and also highlight the performance of other models where they may add value to the overall assessment.

Reviewer #2: Thank you authors for this work that demonstrates the application of machine learning techniques to predict individuals with TB drug non-adherence.

Please find below comments to help improve your manuscript:

Introduction: I find this too long – provide rationale for predicting patients who are likely to be non-adherent. The sentence starting on line 58, "Treatment non-adherence...." breaks the flow with the next paragraph. Please revise to improve on the flow.

Attributes included in development of model should reflect intended public health benefits of model. By the 6th month of treatment, there are a number of missed opportunities to address adherence issues, therefore including sputum collection at 6 months as a predictor of non-adherence is difficult to justify

How did you measure the primary exposure non-adherence? TB patients on treatment do not typically have facility files where self-adherence to drugs is documented. The assumption therefore is that you measured this from records in the Unit TB register. How did you approach this?

In 1 - 2 sentences, describe Mukono district - location, catchment area to provide context to the challenges faced by TB patients in seeking care

Study period overlaps with the COVID19 pandemic lockdown period. Could this have affected your model?

Lines 479 – 481: Include important factors associated with non-adherence

Reference for line 483 – 4 , 487 – 492, 517 – 8, 519 - 21

Was there an attempt to stratify patients by HIV status? CPT status is only applicable to HIV positive individuals – therefore including in model building that involves HIV negative individuals is not proper

Line 537, the sentence starting, “From our findings, ….,” does not build on the previous one that starts, “Comorbidity of tuberculosis and HIV/AIDS………….”. Please revise

Conclusion: Line 549 – The conclusion that drug non-adherence is prevalent among TB patients is not supported by study findings

Is the second paragraph necessary?

Study limitations should be included in the discussion section, before the conclusion. The

6. PLOS authors have the option to publish the peer review history of their article (what does this mean?). If published, this will include your full peer review and any attached files.

**Do you want your identity to be public for this peer review?** For information about this choice, including consent withdrawal, please see our Privacy Policy.

Reviewer #1: **Yes: **Ronald M. Gobina

Reviewer #2: No

---

## [Decision Letter · Decision Letter 1]

24 Apr 2023

PGPH-D-22-01989R1

A machine learning approach to explore individual risk factors for tuberculosis treatment non-adherence in Mukono district

Dear Haron,

Thank you for submitting your manuscript to PLOS Global Public Health. After careful consideration, we feel that it has merit but does not fully meet PLOS Global Public Health’s publication criteria as it currently stands. Therefore, we invite you to submit a revised version of the manuscript that addresses the points raised during the review process.

We look forward to receiving your revised manuscript.

Kind regards,

Collins Otieno Asweto, PhD

Academic Editor

Journal Requirements:

Reviewers' comments:

Reviewer's Responses to Questions

**Comments to the Author**

1. If the authors have adequately addressed your comments raised in a previous round of review and you feel that this manuscript is now acceptable for publication, you may indicate that here to bypass the “Comments to the Author” section, enter your conflict of interest statement in the “Confidential to Editor” section, and submit your "Accept" recommendation.

Reviewer #2: (No Response)

Reviewer #3: All comments have been addressed

2. Does this manuscript meet PLOS Global Public Health’s publication criteria? Is the manuscript technically sound, and do the data support the conclusions? The manuscript must describe methodologically and ethically rigorous research with conclusions that are appropriately drawn based on the data presented.

Reviewer #2: Partly

Reviewer #3: Yes

3. Has the statistical analysis been performed appropriately and rigorously?

Reviewer #2: Yes

Reviewer #3: Yes

4. Have the authors made all data underlying the findings in their manuscript fully available (please refer to the Data Availability Statement at the start of the manuscript PDF file)?

Reviewer #2: Yes

Reviewer #3: Yes

5. Is the manuscript presented in an intelligible fashion and written in standard English?

Reviewer #2: Yes

Reviewer #3: Yes

6. Review Comments to the Author

Reviewer #2: Thank you authors for revising the manuscript. Please find below additional comments:

(a) I note that narratives that best fit in the discussion section still appear in your introduction. Please rectify. For example, lines 123-127 compare your study findings to previous those of studies - this should be presented in the discussion and not the introduction.

(b) The sentence starting at line 50, "Furthermore...." is out of sync with the previous sentence that leads a presentation of factors associated with non-adherence. This paragraph could be better placed before the one presenting these factors, using an appropriate connector (such as moreover or yet at the beginning), with a linking sentence that initially points out the importance of studying factors associated with non-adherence and then leading into a sentence presenting some of the commonly studied factors.

(c) Line 130, the sentence starting, "Unfortunately, based on our research..." should be re-written to, "Unfortunately, based on our literature search...". This is for the purpose of clarifying that the ideas being presented in this sentence are derived from a review of literature and not from the present study findings.

(d) Lines 86 - 136 are based summarised and used to enrich the discussion section. They do not fit well in the introduction.

(e) Line 133: clarify which study is being referred to, in the sentence beginning, "This study...".

(f) Lines 99 - 105: which investigators are being referred to?

(g) Lines 168 - 175 in data description are explanatory. Do we need them here?

(h) Revise and summarise paragraph starting at line 172.

(i) What was the reason for excluding 162 records? Were they duplicates? A clear study flow diagram can help clarify.

(k) Lines 184 -188: How did the additional column in the register facilitate elimination of duplicates? This may be unnecessary detail

(l) Line 266: note and revise hanging sentence

(m) Lines 411 - 414 are misplaced. The first paragraph of the Discussion section should present a summary of the study results, yet these lines are explaining choice of metrics used (this should be moved lower in the discussion section). Lines 429-430 need to be moved to the 1st paragraph of the Discussion. You could refer to this resource for further insights: https://plos.org/resource/how-to-write-conclusions/

(n) Thank you for including the study limitations. You could re-write these so that they don't undermine the credibility of your work.

Reviewer #3: I find this topic very appropriate as it is consistent with the Sustainable Development Goal 3.3. The topic touched on one of the major diseases of public health importance in Africa.

7. PLOS authors have the option to publish the peer review history of their article (what does this mean?). If published, this will include your full peer review and any attached files.

**Do you want your identity to be public for this peer review?** For information about this choice, including consent withdrawal, please see our Privacy Policy.

Reviewer #2: No

Reviewer #3: **Yes: **BOTHA, Nkosi Nkosi

---

## [Decision Letter · Decision Letter 2]

6 Jun 2023

A machine learning approach to explore individual risk factors for tuberculosis treatment non-adherence in Mukono district

PGPH-D-22-01989R2

Dear Gichuhi,

We are pleased to inform you that your manuscript 'A machine learning approach to explore individual risk factors for tuberculosis treatment non-adherence in Mukono district' has been provisionally accepted for publication in PLOS Global Public Health.

Best regards,

Collins Otieno Asweto, PhD

Academic Editor

Reviewer Comments (if any, and for reference):

Reviewer's Responses to Questions

**Comments to the Author**

1. If the authors have adequately addressed your comments raised in a previous round of review and you feel that this manuscript is now acceptable for publication, you may indicate that here to bypass the “Comments to the Author” section, enter your conflict of interest statement in the “Confidential to Editor” section, and submit your "Accept" recommendation.

Reviewer #2: (No Response)

Reviewer #3: All comments have been addressed

2. Does this manuscript meet PLOS Global Public Health’s publication criteria? Is the manuscript technically sound, and do the data support the conclusions? The manuscript must describe methodologically and ethically rigorous research with conclusions that are appropriately drawn based on the data presented.

Reviewer #2: Yes

Reviewer #3: Yes

3. Has the statistical analysis been performed appropriately and rigorously?

Reviewer #2: Yes

Reviewer #3: Yes

4. Have the authors made all data underlying the findings in their manuscript fully available (please refer to the Data Availability Statement at the start of the manuscript PDF file)?

Reviewer #2: Yes

Reviewer #3: (No Response)

5. Is the manuscript presented in an intelligible fashion and written in standard English?

Reviewer #2: Yes

Reviewer #3: Yes

6. Review Comments to the Author

Reviewer #2: Thanks to the authors for an improved manuscript. Final comments from me:

Lines 456-8: the statement, "These results are like other studies conducted in lower- and middle-income countries exploring the patient characteristics associated with treatment non-adherence" should be supported with appropriate references

Line 405: For the opening sentence of the Discussion, I recommend that you maintain, "Our study focused on developing and evaluating different machine learning algorithms to explore the risk factors predictive of treatment non-adherence" then continue to, "Our support vector machine.........."

Line 737 - delete redundant reference number 81

Reviewer #3: All necessary corrections have been addressed.

7. PLOS authors have the option to publish the peer review history of their article (what does this mean?). If published, this will include your full peer review and any attached files.

**Do you want your identity to be public for this peer review?** For information about this choice, including consent withdrawal, please see our Privacy Policy.

Reviewer #2: No

Reviewer #3: **Yes: **BOTHA, Nkosi Nkosi
